# Impact of Pancreatitis-Associated Protein on Newborn Screening Outcomes and Detection of CFTR-Related Metabolic Syndrome (CRMS)/Cystic Fibrosis Screen Positive, Inconclusive Diagnosis (CFSPID): A Monocentric Prospective Pilot Experience

**DOI:** 10.3390/ijns8030046

**Published:** 2022-08-03

**Authors:** Chiara Bianchimani, Daniela Dolce, Claudia Centrone, Silvia Campana, Novella Ravenni, Tommaso Orioli, Erica Camera, Gianfranco Mergni, Cristina Fevola, Paolo Bonomi, Giovanni Taccetti, Vito Terlizzi

**Affiliations:** 1Cystic Fibrosis Regional Reference Center, Department of Paediatric Medicine, Meyer Children’s Hospital, 50139 Florence, Italy; bianchimanichiara@gmail.com (C.B.); daniela.dolce@meyer.it (D.D.); silva.campana@meyer.it (S.C.); novella.ravenni@meyer.it (N.R.); tommaso.orioli@meye.it (T.O.); erica.camera@meyer.it (E.C.); gianfranco.mergni@meyer.it (G.M.); cristina.fevola@meyer.it (C.F.);giovanni.taccetti@meyer.it (G.T.); 2Diagnostic Genetics Unit, Careggi University Hospital, 50139 Florence, Italy; centronec@aou-careggi.toscana.it; 3Freelance Statistician, 20100 Milan, Italy; paolo_bonomi@outlook.it

**Keywords:** PAP, screening, outcomes, PPV, sweat chloride

## Abstract

Pancreatitis-Associated Protein (PAP)-based Cystic Fibrosis (CF) newborn bloodspot screening (NBS) protocols detect less CFTR-Related Metabolic Syndrome (CRMS)/CF Screen Positive, Inconclusive Diagnosis (CFSPID). We prospectively evaluated the impact of PAP as the second step of the CF NBS protocol, before the CFTR genetic analysis, on NBS outcomes and CRMS/CFSPID detection in the Tuscany region, Italy. In parallel to the usual protocol (IRT/DNA, protocol 1), PAP was analyzed in IRT-positive infants (IRT/PAP/DNA, protocol 2) from 1 June 2020 until 31 May 2022. We defined an infant as NBS positive if PAP was >1.8 μg/L for IRT value 99th percentile-100 μg/L or >0.6 μg/L for IRT value >100 μg/L. To increase the positive predictive value (PPV) of protocol 2, we retrospectively lowered the upper IRT range value from 100 to 90 μg/L (modified protocol 2). We identified 8 CF and 13 CRMS/CFSPID with protocol 1, 5 CF and 5 CRMS/CFSPID with protocol 2 and 8 CF and 5 CRMS/CFSPID with modified protocol 2. With the PAP-based protocols, we observed a reduction of sweat tests, healthy carrier detection and a significant increase in PPV to 15.38%. Further data are needed in order to evaluate the outcomes of CRMS/CFSPID after a long follow-up.

## 1. Introduction

Cystic fibrosis (CF) is the most common life-limiting autosomal recessive disease in Caucasian populations due to variants in the CF Transmembrane Conductance Regulator (CFTR) gene. The CF phenotype is characterized by lung disease, exocrine pancreatic insufficiency associated with nutrient malabsorption contributing to undernutrition, impaired growth, hepatobiliary manifestations, and male infertility [1]. Newborn bloodspot screening (NBS) for CF is a well-established public health strategy based on international standards. The goal of CF NBS is to achieve an early CF diagnosis so that comprehensive medical and psychosocial therapies can be implemented in infants prior to the onset of clinical symptoms. CF NBS, when associated with early treatment, limits lung damage in childhood, has a beneficial effect on clinical outcomes, reduces the burden of care for families, and may improve survival [2,3,4].

All NBS programs start by measuring the concentration of immunoreactive trypsinogen (IRT) in dried blood spots. The second tier is either a limited CFTR variant analysis or a repeat measurement of the IRT concentration at the age of 4–6 weeks [5]. Further outcomes, after CFTR genetic analysis introduction include the identification of carrier status [6] and the emergence of a cohort of infants with positive NBS test results but an inconclusive diagnosis, classified as having CF transmembrane conductance regulator-related metabolic syndrome (CRMS)/CF screen–positive, inconclusive diagnosis (CFSPID) [7,8].

A percentage of these subjects will remain healthy in most cases but can receive a CF diagnosis over time due to a positive sweat test or a re-classification of CFTR variants as CF causing or rarely leading to the development of clinical CF features [9,10,11,12]. The prevalence of CRMS/CFSPID cases is highly variable and depends on the specific NBS algorithm, increasing for protocols that use DNA analysis and even more CFTR gene sequencing [5,9,12]. However, CRMS/CFSPID is an unintended consequence of CF NBS, causing long-term psychosocial, medical and financial impacts. For these reasons, reducing the number of CRMS/CFSPID would be desirable. In 2005, pancreatitis-associated protein (PAP) was described as a possible second tier in NBS for CF [13]. 

PAP is a secretory protein that is not measurable in blood under normal conditions but which can be detected in high quantities in the context of pancreatic injury [14,15]. PAP may already be synthesized in utero in the CF pancreas and present in blood at birth [16]. In the first French pilot studies, it was found that newborns with CF always had both an increased IRT and an increased PAP, whereas those without CF showed elevation of IRT or PAP, but rarely of both [13,15,16]. Subsequently, PAP was inserted as a second tier in other NBS protocols, such as in the Netherlands and Portugal [5,17,18], and more recently, the outcomes have also been reported in several other countries [19,20,21,22,23]. The IRT/PAP strategy avoids the drawbacks of genetic analysis and is cheaper and easier to implement than the current IRT/CFTR mutation strategy. Furthermore, PAP-based protocols have advantages in multi-ethnic populations, improve the positive predictive value (PPV), and help to detect fewer carriers and CRMS/CFSPID [15].

The aim of this paper was to evaluate the impact of PAP as the second tier of CF NBS protocol, before the CFTR genetic analysis, on NBS outcomes and on detection of CRMS/CFSPID in an Italian region with a high prevalence of CRMS/CFSPID. 

## 2. Materials and Methods

### 2.1. Study Population and NBS Protocols

This is a prospective pilot study including all CF NBS positive subjects born in the Tuscany region, Italy, from 1 June 2020 until 31 May 2022.

The protocol used was IRT/DNA (protocol 1): IRT was measured for all newborns from a blood spot sample taken on the third day of life using the GSP instrument (Perkin-Elmer, Waltham, MA, USA). The IRT cut-off value > 99th percentile, calculated every four months, was in the range of 47–50 ng/mL. All infants with an IRT > 99th percentile performed CFTR genetic analysis, including all CFTR-causing variants, according to the CFTR2 database (https://cftr2.org/, accessed on 20 June 2022). The complete algorithm is described in Figure 1.

In parallel to protocol 1, PAP was performed in newborns with IRT > 99th percentile (IRT/PAP/DNA, protocol 2). PAP was assayed from the same screening card using MUCOPAP-F kit (Dynabio, France), a Time-resolved fluoro-immunoassay (TRF-IA). We defined an infant as CF NBS positive if PAP was >1.8 μg/L for IRT value between 99th percentile and 100 μg/L or >0.6 μg/L for IRT value >100 μg/L [24]. The complete algorithm is described in Figure 2.

We excluded from the analysis cases in which the sweat test was performed in the presence of meconium ileus since it is a typical symptom of CF and all infants who received a transfusion at birth.

We defined a subject as CRMS/CFSPID or CF according to guidelines [7,25]. Sweat chloride (SC) levels were tested according to guidelines [26] and performed only in the laboratory of the CF center of Florence by expert operators, thus ruling out the lack of harmonization [27].

All CRMS/CFSPID underwent 2 SC tests on the first day, and the SC test was repeated every 6 months until at least two negative or pathologic values appeared or until 31 May 2022, in case of persistent intermediate SC values [11,28]. We carried out gene sequencing (detection rate 98%) and multiplex ligation-dependent probe amplification (MLPA) in all CRMS/CFSPID infants in whom one variant was found after first-level analysis.

During follow-up, we reclassified CRMS/CFSPID babies as CF diagnosis, healthy carrier or healthy subject, as already reported [12]. A CRMS/CFSPID label was kept in infants with SC levels persistently in the intermediate range or in the presence of 2 CFTR variants, at least 1 of which had unclear phenotypic consequences (https://www.cftr2.org/, accessed on 20 June 2022).

The study was approved by the local ethics committee (Meyer Children’s Hospital, number 55/2020), and informed written consent was obtained from the parents of involved subjects for the use of anonymous clinical data for research purposes.

### 2.2. Statistics

Descriptive statistics for continuous variables were obtained according to normal distribution tests. The diagnostic performances (sensitivity, specificity, PPV, negative predictive value (NPV), likelihood ratio positive (LR+), likelihood ratio negative (LR-) values), and their 95% confidence limits were calculated for the two algorithms individually, comparing CF diagnosis vs. no CF (CRMS/CFSPID + healthy). Comparisons between independent samples were performed using Fisher’s exact test. All tests were considered significant at the level of *p* < 0.05.

## 3. Results

Of 45,862 babies screened from 1 June 2020 to 31 May 2022, 446 (0.01%) resulted IRT-positive (≥99th percentile) at day 3. One hundred twenty-seven (28.5%) out of 446 had a pathological PAP value: 107 (84.3%) had PAP > 1.8 μg/L and IRT in the range 99th percentile and 100 μg/L, while 20 (15.7%) out of 127 had PAP > 0.6 μg/L and IRT > 100 μg/L.

We identified:

-8 CF diagnosis and 13 CRMS/CFSPID with protocol 1;

-5 CF diagnosis and 5 CRMS/CFSPID with protocol 2, according to the cut-offs of Sarles et al. [24].

All CRMS/CFSPID had a SC in intermediate range (30–59 mEq/L) (Table 1).

In order to increase the PPV of protocol 2 and identify all CF patients, we retrospectively changed the cut-offs as follows: Positive NBS for PAP > 1.8 μg/L and IRT between 99th percentile value and 90 μg/L or PAP > 0.6 μg/L and IRT value > 90 μg/L (we called it “modified protocol 2”). One hundred thirty (29.1%) out of 446 had a pathological PAP value: 104 (80%) had PAP > 1.8 μg/L and IRT in the range 99th percentile and 90 μg/L, while 26 (20%) out of 130 had PAP > 0.6 μg/L and IRT > 90 μg/L. In this way, we identified all 8 CF diagnosis and the same 5 CRMS/CFSPID (Table 1). Therefore, with the modified protocol 2, we found a reduction in 38.5% in CRMS/CFSPID detection, not identifying 8 CRMS/CFSPID, with a significant increase of PPV until 15.38% (*p* 0.033, Table 2).

Furthermore, we identified 59 health carriers with protocol 1, 16 with protocol 2 and 16 with modified protocol 2.

152 newborns needed SC test with protocol 1, 48 with protocol 2, and 52 with modified protocol 2, so with a reduction of 31.6% and 34.2% of sweat test, respectively.

We highlight a difference (t (7.2) = −1.69, *p* = 0.133) in PAP values between the CF patients (n.8, mean: 5.45 SD: 6.51) and CRMS/CFSPID (n.13, mean: 1.52 SD: 1.07), standard deviations are not equal (Snedecor Test: F = 18.7 *p* < 0.001). Furthermore, IRT values were significantly higher in CF patients than in CRMS/CFSPID (mean 105.13 vs. 64.62; SD: 37.67 vs. 22.33, respectively, t (19) = −3.1, *p* < 0.05).

Finally, in Table 3, we report the IRT and PAP values also in healthy subjects and healthy carriers.

After a mean follow-up of 12 months (range 6–23 months), 5 out of 13 CRMS/CFSPID became healthy carriers at a mean age of 8.8 months, while 8 kept a CRMS/CFSPID label: Of these, 4 in the presence of SC in the intermediate range and 4 with normal SC and a second CFTR non causing variant at CFTR2 database (L997F, F508C) or a rare variant (M952I) not reported at CFTR2 database.

## 4. Discussion

In this pilot study, we evaluated the potential impact of the introduction of PAP as the second tier in NBS protocol, before CFTR genetic analysis, in reducing the number of CRMS/CFSPID and increasing the PPV.

It is known that the prevalence of CRMS/CFSPID is primarily influenced by the CF NBS algorithms. In the Italian population, it is very high [12,28,29,30], thus screening programs could benefit from the use of PAP. This may be due to several aspects: The prevalence of F508del is lower than in American, Australian, or Northern European population. Furthermore, the main regional algorithms use a very large first-level CFTR genetic panel, with subsequent genetic sequencing, identifying many CRMS/CFSPID with unknown variants and sweat tests normal or with variants with variable clinical consequences [12]. Furthermore, in our previous study we showed that 17 (34%) and 15 (30%) out of 50 CRMS/CFSPID followed at CF centre of Florence and born in 2011–2016 were healthy subjects or healthy carriers [28]. For these reasons, reducing the number of CRMS/CFSPID was our priority to reduce false positives and parental anxiety.

To date, there are no PAP cut-offs used in Italian subjects. As in previous studies, first of all, we used the PAP value used for French populations [24]. In this way, we identified fewer CRMS/CFSPID, but we noted a low PPV (10.42%), not identifying three CF diagnosis, compared to the usual protocol IRT-DNA. The CF French cohorts are generally quite different from the Italian ones, probably due to a multi-ethnic population with a higher prevalence of non-Caucasian subjects and a higher prevalence of CFTR variants, such as R117H [10,31], which are rarely found in Italy [12,28]. Subsequently, we retrospectively modified the IRT cut-offs and this change resulted in a 38.5% reduction in CRMS/CFSPID findings and a significant increase in PPV up to 15.38%, identifying all new CF diagnosis. Furthermore, the algorithm IRT-PAP-DNA allowed to reduce the number of sweat chloride tests conducted, an important source of anxiety for families and the number of healthy carriers, another unwanted outcome of CF NBS.

Of the PAP-based CF NBS protocols currently used in a national or regional screening program, only the Netherlands has so far reached the ECFS recommended PPV, ie > 30% [17]. This performance was achieved with the protocol IRT-PAP-DNA analysis (including 35 CFTR variants)—extended gene analysis as the fourth step and as a safety net. In all blood spots with IRT concentrations ≥ 60 μg/L blood, PAP concentrations were measured. When PAP was ≥3.0 μg/L blood, or PAP ≥ 1.6 μg/L blood, and IRT ≥ 100 μg/L blood, DNA analysis was performed [17]. A PAP-based protocol with a DNA analysis as the third tier was also used in Germany, Catalonia, Belgium, and the Czech Republic, using variable PAP cut-off values and searching for the most common disease-causing CFTR variants [15,22,23,32]. Differently from other cohorts, we searched all CFTR-causing variants in the CFTR2 database in newborns with IRT and PAP positive. Furthermore, as reported in methods, PAP was assayed from the same screening card used for the IRT assay, so it was taken on the third day of life and in no case before 48 h of life, that is when PAP value is not sensitive enough [15]. Nevertheless, we observed an improvement in PPV value far from the recommended value of 30%. Anyway, the PPV value is strongly influenced by disease incidence in contrast to sensitivity and specificity.

So far, there is no evidence that the PAP values correlate with the severity of CF disease. It is known that higher PAP values are in CRMS/CFSPID or CF patients with CFTR variants leading to pancreatic sufficiency, and low PAP values are in some patients with CFTR variants leading to pancreatic insufficiency and a severe CF phenotype [33]. In our cohort, CRMS/CFSPID had similar PAP values compared to healthy carriers or healthy subjects (Table 3). On the other hand, CF newborns had a higher PAP value (although a non-significant difference probably due to the small number of patients), especially two CF patients with pancreatic insufficiency (number 2 and 6 of Table 2). Obviously, our cohort is too small to draw conclusions, and more data are needed.

With protocol 1 (IRT/DNA), we identified 13 CRMS/CFSPID, and of these, five (38.5%) had a final diagnosis as healthy carriers at 8.8 months. These data confirm the usefulness of repeating the sweat test every six months in the first years of life, in order to identify non-CF subjects early and interrupt the follow-up as soon as possible [34].

Protocol 2 (IRT/PAP/DNA) did not identify eight CRMS/CFSPID: Two with a final diagnosis of healthy carriers (Table 1) and four with normal sweat chloride on 31 May 2022 and a second not CF-causing variant at CFTR2 (L997F and F508C) or reported only in CFTR1 database (M952I) in two individuals, with CBAVD or pancreatic insufficiency and pathological SC value. These individuals likely have a very low risk of progressing to CF, but we cannot exclude that during the follow-up, these children develop a mono-organ involvement for CFTR-related disorder. Furthermore, the two remaining CRMS/CFSPID carried a genetic profile F508del/5T;TG12 and F508del/S737F, both found in Italian CRMS/CFSPID progressed to CF [35,36]. In fact, we recently reported that, after a median follow-up of 6.7 years, 10.3% of CRMS/CFSPID subjects with F508del/5T;12TG genotype progressed to CF, which was a higher percentage than that previously reported in a large cohort of Italian CRMS/CFSPID [35]. Finally, S737F is a CFTR variant typical of the Tuscany region and associated with CF evolution [36]. For these reasons, a longer follow-up is necessary to evaluate the possible evolution in CF of these two subjects not identified by protocol 2. The short follow-up is an important limitation of our paper. We do not know the number of false negative subjects and the evolution of unidentified CRMS/CFSPID subjects. Furthermore, we have adapted a cut-off of the PAP value that was not appropriate for the Italian population, modifying it retrospectively in order to identify all CF subjects.

## 5. Conclusions

In conclusion, our data show that the insertion of PAP as a second tier of NBS protocol increases the positive predictive value, selecting newborns for genetic analysis and reducing costs. However, further data are needed in order to establish the outcomes of CRMS/CFSPID after a long follow-up.

The PAP cut-off is dependent on several factors and needs to be adapted for the specific cohort. The modified PAP value used in our cohort could be valid for the Italian population.

## Figures and Tables

**Figure 1 IJNS-08-00046-f001:**
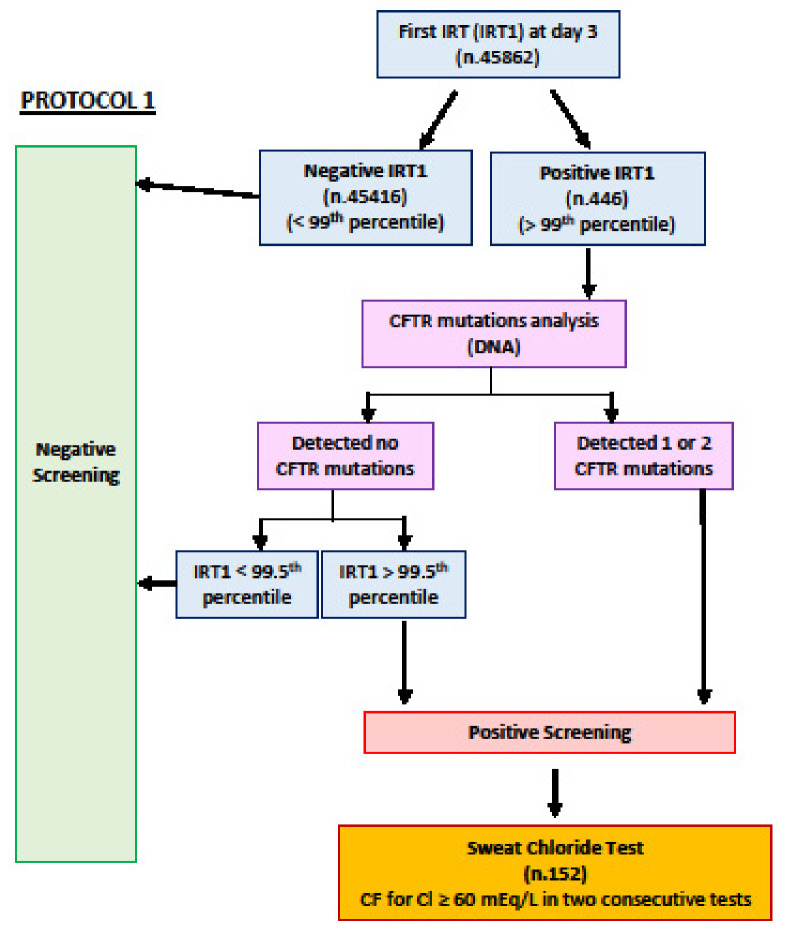
Protocol 1 IRT/DNA.

**Figure 2 IJNS-08-00046-f002:**
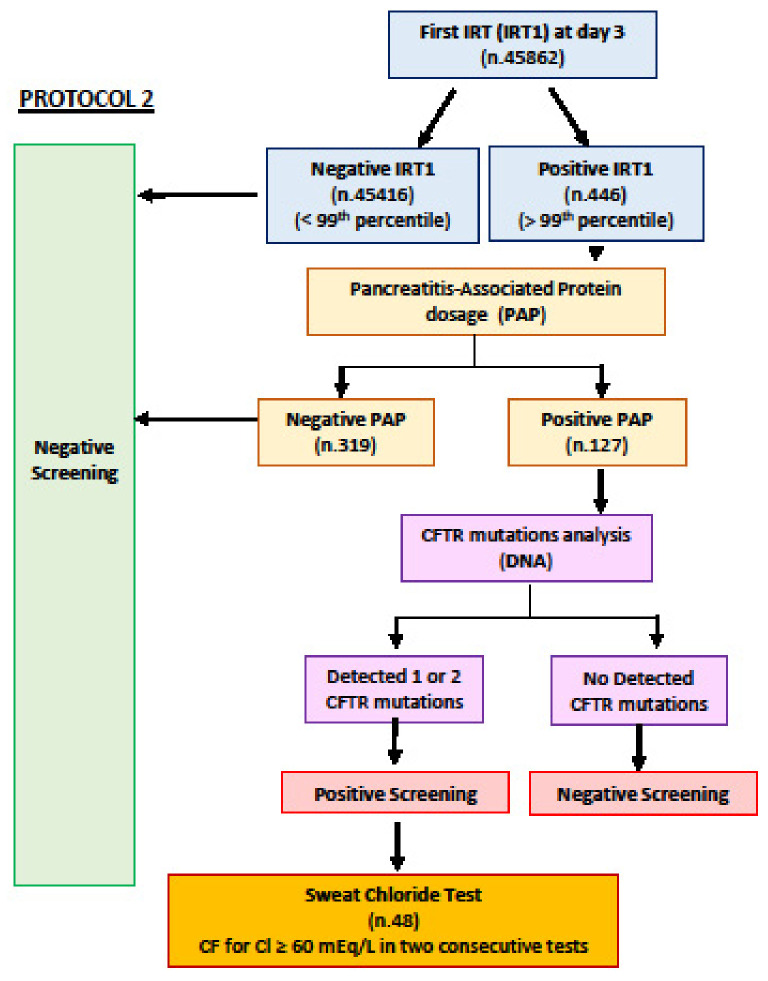
Protocol 2 (IRT/PAP/DNA).

**Table 1 IJNS-08-00046-t001:** Genotypes; IRT, PAP, and sweat chloride values and final diagnosis of subjects with CRMS/CFSPID and CF diagnosis.

		IRT (ng/mL)	PAP (μg/L)	CFTR Variant 1	CFTR Variant 2	First SC (mEq/L)	Last SC (mEq/L), Age (Months)	Diagnosis at Study End (31 May 2022)
CF								
	1	98	1.53	N1303K	N1303K	126	/	CF
	2	75	13.95	L867X	G378X/I148T ^a^	109	/	CF
	3	91	1.13	F508del	G126D	66	/	CF
	4	182	0.86	F508del	F508del	84	/	CF
	5	87	4.70	F508del	c.870-113_870-1110delGAAT	65	/	CF
	6	141	17.44	F508del	F508del	126	/	CF
	7	96	1.24	R117C	G542X	48	51 (2)	CF
	8	71	2.76	R553X	2789+5G-> A	93		CF
CRMS/CFSPID								
	1 *	93	2.07	E585X	UN ^c^	49	16 (18)	healthy carrier
	2	47	0.60	F508del	5T; TG12	50	45 (11)	CMRS/CFSPID
	3	53	1.18	F1052V/621+3° > G ^a^	UN ^c^	32	18 (6)	healthy carrier
	4	54	0.27	F508del	L997F ^b^	36	29 (21)	CRMS/CFSPID
	5 *	58	3.01	F508del	S912L	36	48 (19)	CRMS/CFSPID
	6	58	1.11	N1303K	F508C ^b^	31	25 (19)	CRMS/CFSPID
	7 *	129	3.02	2789+5G -> A	5T-12TG	39	34 (19)	CRMS/CFSPID
	8 *	54	3.20	F508del/L467P ^a^	UN ^c^	31	31 (9)	healthy carrier
	9 *	54	2.01	F508del	UN ^c^	49	18(8)	healthy carrier
	10	65	1.25	F508del	S737F	51	50 (9)	CRMS/CFSPID
	11	53	1.01	L1065P	L997F ^b^	31	18 (8)	CRMS/CFSPID
	12	60	0.61	F508del	UN ^c^	32	25 (6)	healthy carrier
	13	62	1.58	D110H	M952I ^d^	32	17 (6)	CRMS/CFSPID

Abbreviations: CRMS/CFSPID: CFTR-Related Metabolic Syndrome/CF Screen Positive, Inconclusive Diagnosis; CF: Cystic Fibrosis; PAP: Pancreatitis-Associated Protein; IRT: Immunoreactive trypsinogen; SC: Sweat chloride; UN: Undetected; N/A: Not available (quantity of sweat < 75 mg); >max: Value beyond the upper limit of detection; * CRMS/CFSPID identified with both protocols; ^a^ complex allele; ^b^ not causing variant at CFTR2 database; ^c^ after gene sequencing (detection rate 98%); ^d^ CFTR variant reported only at CFTR1 database (http://www.genet.sickkids.on.ca/, accessed on 20 June 2022).

**Table 2 IJNS-08-00046-t002:** Diagnostic performances of two newborn screening algorithms (estimated value and confidence interval between parentheses).

			CF vs. (CRMS/CFSPID + Healthy)		
	IRT-DNA Protocol 1	IRT-PAP-DNA Protocol 2	IRT-PAP-DNA Modified Protocol 2	*p* Value Protocol 1 vs. Protocol 2	*p* Value Protocol 1 vs. Modified Protocol 2
Sensitivity %	100.00 (63.06–100.00)	100.00 (47.82–100.00)	100.00 (63.06–100.00)		
Specificity %	99.69 (99.64–99.74)	99.91 (99.87–99.93)	99.90 (99.87–99.93)	<0.001	<0.001
PPV %	5.33 (4.56–6.23)	10.42 (7.94–13.55)	15.38 (11.92–19.63)	0.311	0.033
NPV %	100.00	100.00	100.00		
Positive LR	322.92 (274.01–380.55)	1066.44 (791.02–1437.76)	1042.16 (775.66–1400.23)	<0.001	<0.001
Negative LR	0.00	0.00	0.00		

Abbreviations: PPV: Positive predictive value; NPV: Negative predictive value; LR = Likelihood ratios, IRT1: Immunoreactive trypsin, CRMS/CFSPID: CFTR-Related Metabolic Syndrome/CF Screen Positive, Inconclusive Diagnosis; CF: Cystic Fibrosis; PAP: Pancreatitis-Associated Protein; IRT: Immunoreactive trypsinogen.

**Table 3 IJNS-08-00046-t003:** Mean IRT and PAP value in healthy subjects, healthy carriers, CRMS/CFSPID, and CF patients.

	IRT		PAP	
Screening Diagnosis (Newborns Number)	Mean (ng/mL)	SD	Mean (μg/L)	SD
Healthy (366)	61.17	38.86	1.86	5.97
Healthy carriers (59)	59.97	16.77	1.39	0.96
CRMS/CFSPID (13)	64.62	22.33	1.52	1.07
CF (8)	105.13	37.67	5.45	6.51

Abbreviations: CRMS/CFSPID: CFTR-Related Metabolic Syndrome/CF Screen Positive, Inconclusive Diagnosis; CF: Cystic Fibrosis; PAP: Pancreatitis-Associated Protein; IRT: Immunoreactive trypsinogen.

## Data Availability

All data are available by contacting corresponding author (V.T).

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
