# Peer review of "Impact of Pancreatitis-Associated Protein on Newborn Screening Outcomes and Detection of CFTR-Related Metabolic Syndrome (CRMS)/Cystic Fibrosis Screen Positive, Inconclusive Diagnosis (CFSPID): A Monocentric Prospective Pilot Experience"

_2409-515X, 2022, doi:10.3390/ijns8030046_

Round 1

Reviewer 1 Report

Well-written paper, easy to read and with relevant results. In Figure 2 we can assume that if no CFTR mutations are detected, the screening will be negative but it could be more clear if this information was added to the figure.

The indication in figures 1 and 2 of the number of newborns identified in each step could also help the interpretation of the figures.

Author Response

Impact of Pancreatitis-Associated Protein on newborn screening outcomes and detection of CFTR-Related Metabolic Syndrome (CRMS)/Cystic Fibrosis Screen Positive, Inconclusive Diagnosis (CFSPID): a monocentric prospective pilot experience

Bianchimani C et al.

Reviewer 1

Well-written paper, easy to read and with relevant results.

Re: we thank the reviewer for the relevant comments and for taking the time to review our paper.

In Figure 2 we can assume that if no CFTR mutations are detected, the screening will be negative but it could be more clear if this information was added to the figure.

Re: we have added the indicated information in Figure 2.

The indication in figures 1 and 2 of the number of newborns identified in each step could also help the interpretation of the figures.

Re: we have added the indicated information in Figure 1 and 2.

Reviewer 2 Report

This is a well written paper reporting a well-designed study employing a sequential quality improvement strategy taking advantage of the ancillary role of PAP in CF newborn screening.

Some revisions in the Discussion would strengthen the manuscript, in my judgment. My recommendations are listed below.

In the Discussion (page 8, lines 213-214), the authors should recognize that the statement It’s known that the prevalence of CRMS/CFSPID is very high in the Italian population is not appropriate even it appears to be true. As many studies show and this study confirms, the prevalence of CRMS/CFSPID is influence primarily by the CF newborn screening algorithm.

The PPV issue should be discussed more critically. It’s clear that the ECFS recommended PPV, ie > 30% is unrealistic. PPV is strongly influenced by disease incidence in contrast to sensitivity and specificity. Thus, paragraph #4 of the Discussion should be rewritten.

In addition to cutoff value issues, there are other problems with PAP that should be discussed such as timing in relationship to birth and heat lability. Please discuss these.

Author Response

Impact of Pancreatitis-Associated Protein on newborn screening outcomes and detection of CFTR-Related Metabolic Syndrome (CRMS)/Cystic Fibrosis Screen Positive, Inconclusive Diagnosis (CFSPID): a monocentric prospective pilot experience

Bianchimani C et al.

Reviewer 2

This is a well written paper reporting a well-designed study employing a sequential quality improvement strategy taking advantage of the ancillary role of PAP in CF newborn screening.

Some revisions in the Discussion would strengthen the manuscript, in my judgment. My recommendations are listed below.

Re: we thank the reviewer for the relevant comments and for taking the time to review our paper.

In the Discussion (page 8, lines 213-214), the authors should recognize that the statement It’s known that the prevalence of CRMS/CFSPID is very high in the Italian population is not appropriate even it appears to be true. As many studies show and this study confirms, the prevalence of CRMS/CFSPID is influence primarily by the CF newborn screening algorithm.

Re: we have modified the sentence.

The PPV issue should be discussed more critically. It’s clear that the ECFS recommended PPV, ie > 30% is unrealistic. PPV is strongly influenced by disease incidence in contrast to sensitivity and specificity. Thus, paragraph #4 of the Discussion should be rewritten.

Re: we have modified the sentence as suggested.

In addition to cutoff value issues, there are other problems with PAP that should be discussed such as timing in relationship to birth and heat lability. Please discuss these.

Re: we believe these points could be confusing. We had no difficulty in this.
